# Cajanolactone A from *Cajanus cajan* Promoted Osteoblast Differentiation in Human Bone Marrow Mesenchymal Stem Cells via Stimulating Wnt/LRP5/β-Catenin Signaling

**DOI:** 10.3390/molecules24020271

**Published:** 2019-01-12

**Authors:** Shan Liu, Zhuo-Hui Luo, Gui-Mei Ji, Wei Guo, Jia-Zhong Cai, Lin-Chun Fu, Juan Zhou, Ying-Jie Hu, Xiao-Ling Shen

**Affiliations:** 1Laboratory of Chinese Herbal Drug Discovery, Institute of Tropical Medicine, Guangzhou University of Chinese Medicine, Guangzhou 510405, Guangdong, China; liushan@gzucm.edu.cn (S.L.); zhuohuiluo@126.com (Z.-H.L.); 17620027119@163.com (G.-M.J.); 18825099643@163.com (W.G.); linchunfu@gzucm.edu.cn (L.-C.F.); rysyxs@gzucm.edu.cn (J.Z.); 2The Research Center of Integrative Medicine, School of Basic Medical Sciences, Guangzhou University of Chinese Medicine, Guangzhou 510405, Guangdong, China; 3Pi-Wei Institute, Guangzhou University of Chinese Medicine, Guangzhou 510405, Guangdong, China; jiazhongcai@gzucm.edu.cn

**Keywords:** cajanolactone A, *Cajanus cajan* (L.) Millsp., osteoblast differentiation, Wnt/β-catenin signaling pathway

## Abstract

Cajanolactone A (CLA) is a stilbenoid discovered by us from *Cajanus cajan* (L.) Millsp. In our study, CLA was found to promote osteoblast differentiation in human bone marrow mesenchymal stem cells (hBMSCs), as judged by increased cellular alkaline phosphatase activity and extracellular calcium deposits, and elevated protein expression of Runx2, collagen-1, bone morphogenetic protein-2, and osteopontin. Mechanistic studies revealed that hBMSCs undergoing osteoblast differentiation expressed upregulated mRNA levels of *Wnt3a*, *Wnt10b*, *LRP5/6*, *Frizzled 4*, *β-catenin*, *Runx2*, and *Osterix* from the early stage of differentiation, indicating the role of activated Wnt/β-catenin signaling pathway in osteoblast differentiation. Addition of CLA to the differentiation medium further increased the mRNA level of *Wnt3a*, *Wnt10b*, *Frizzled 4*, *LRP5*, and *β-catenin*, inferring that CLA worked by stimulating Wnt/LRP5/β-catenin signaling. Wnt inhibitor dickkopf-1 antagonized CLA-promoted osteoblastogenesis, indicating that CLA did not target the downstream of canonical Wnt signaling pathway. Treatment with CLA caused no changes in mRNA expression level, as well as protein secretion of osteoprotegerin (OPG) and receptor activator of nuclear factor kappa-B ligand (RANKL), indicating that CLA did not affect the OPG/RANKL axis. Our results showed that CLA, which promoted osteoblast differentiation in hBMSCs, through activating Wnt/LRP5/β-catenin signaling transduction, is a promising anti-osteoporotic drug candidate.

## 1. Introduction

Wnt/β-catenin signaling pathway, also known as the canonical Wnt signaling pathway, plays a key role in the differentiation of bone marrow mesenchymal stem cell (BMSC) into osteoblast and regulates osteoblast function, from initial fate determination to its maturation [1]. In the presence of Wnt ligands, Wnt binds to the transmembrane receptor frizzled (Fzd) and co-receptor low density lipoprotein receptor-related protein 5/6 (LRP5/6); Disheveled (Dvl) receives signals from the Fzd/LRP complex, leading to inhibition of GSK3β by phosphorylation; and β-catenin cannot be phosphorylated by GSK3β, thus allowing β-catenin levels to accumulate, which can enter the nucleus and interact with the transcription factor T-cell factor/lymphoid enhancer factor (TCF/LEF) to activate the transcription of Wnt target genes [2,3]. Osteoblast differentiation is negatively regulated by a number of direct Wnt inhibitors, including secreted frizzled related proteins (sFRPs) which bind directly to Wnt ligands and thus prevent the formation of Wnt-Fzd complexes [4], dickkopfs (DKKs), and Sclerostin proteins, which bind to the extracellular domains of LRP5/6 and interfere with their interaction with Wnt proteins [5,6], etc. Targeting Wnt signaling pathway is a feasible approach for the treatment of osteoporosis [7]; the existence of potential pharmacological targets in this pathway makes it abundantly attractive for anti-osteoporotic drug discovery.

*Cajanus cajan* (L.) Millsp. is a traditional herbal medicine in southern China which belongs to perennial shrub of Papilionaceae DAL. Its fresh leaves are used to treat various diseases, including parasitosis [8], swelling [9], oxidative damage [10], and cancer [11]. A water extract of *C. cajan* was reported to promote the proliferation of BMSCs and be beneficial to clinical applications of ischemic necrosis of femoral head [12]. In our previous investigation, a hydrophobic fraction from the leaves of *C. cajan* showed ability to increase tibial bone density in diabetic/obese mice [13]. From this hydrophobic fraction, three new stilbenoids, cajanstilbene H, cajanonic acid A, and cajanolactone A, were discovered by us [14]. Among them, cajanstilbene H exhibited in vitro osteogenesis-promoting activity [15], while cajanonic acid A showed therapeutic potential in the treatment of type 2 diabetes [16]. In this study, we report our new findings that cajanolactone A promoted osteoblast differentiation in human bone marrow mesenchymal stem cells (hBMSCs) via stimulating Wnt/LRP5/β-catenin signaling, and is a promising anti-osteoporotic drug candidate.

## 2. Results and Discussion

### 2.1. Cajanolactone A did not Stimulate the Proliferation of hBMSCs

The effect of cajanolactone A (CLA) (Figure 1A) on proliferation of hBMSCs was firstly investigated. It was found that treatment with CLA for 48 or 72 h did not stimulate cell growth and, at 20 μM, it even exhibited an inhibitory effect on cell proliferation. The measured maximum no-effect concentration of CLA was 10 μM (Figure 1B). In order to avoid CLA-caused growth inhibition, the maximum working concentration of CLA in followed differentiation experiments was set as 4 μM.

### 2.2. CLA Promoted Osteoblast Differentiation in hBMSCs

To investigate the effect of CLA on osteogenesis, hBMSCs were induced for osteoblast differentiation in the presence or absence of CLA (1, 2, 4 µM) for 15 d. Alkaline phosphatase (ALP) activity and calcium deposits (indicators of successful osteoblast differentiation), and protein expression of bone morphogenetic protein-2 (BMP-2), osteopontin (OPN), and collagen-1 (osteoblast-specific proteins) were assessed, and the results are shown in Figure 2. Compared to osteogenic induction without CLA, osteogenic induction with CLA achieved dose-dependent increases in cellular ALP activity (Figure 2A,B) and extracellular calcium deposits (Figure 2C,D). Consistent with this, expression of BMP-2, OPN, and Collagen-1 were also elevated by CLA (Figure 2E,F). The results indicated that CLA promoted the osteoblast differentiation of hBMSCs.

### 2.3. CLA Promoted the Activation of Wnt/β-Catenin Signaling Pathway in hBMSCs

The activation of Wnt/β-catenin signaling pathway plays a key role in osteoblastogenesis. Runx2, the key transcriptional factor for osteoblast differentiation, is the principal downstream target of Wnt/β-catenin signaling, and is activated by β-catenin that enters the nucleus [17]. In this study, hBMSCs at the 1st, 3rd, 6th, 9th, and 15th day of osteoblast differentiation were investigated for the mRNA expression of genes involved in Wnt/β-catenin signaling transduction by real-time quantitative polymerase chain reaction (RT-qPCR). As shown in Figure 3A, hBMSCs undergoing osteoblast differentiation expressed markedly upregulated mRNA level of *Wnt3a*, *Wnt10b*, *LRP5*, *LRP6*, *Frizzled 4*, *β-catenin*, *Runx2*, and *Osterix*, indicating the activation of canonical Wnt signaling pathway from the early stage of differentiation. Results from Western blot analysis and immunohistochemical analysis provided evidence for the activation of Wnt/β-catenin signaling: hBMSCs induced for osteoblast differentiation expressed obviously higher levels of Runx2 than cells that were not induced (Figure 4), and had β-catenin translocated into the nucleus (Figure 5).

Compared to osteoblast differentiation without CLA, the differentiation with CLA (1, 2, or 4 μM) achieved further increases in the mRNA levels of *Wnt3a*, *Wnt10b*, *LRP5*, *Frizzled 4*, *β-catenin*, *Runx2*, and *Osterix* in hBMSCs, but kept the mRNA level of *LRP6* unaffected. Consistent with this, protein expression of β-catenin and Runx2, as well as the nuclear translocation of β-catenin, were distinctly increased by CLA (Figure 3B). The results strongly suggest that CLA promoted osteoblast differentiation via stimulating Wnt/LRP5/β-catenin signaling transduction.

### 2.4. CLA did not Target the Downstream of Canonical Wnt Signaling Pathway

In addition to activation of Wnt ligands, direct inhibition of GSK3β by phosphorylation at Ser9 [18,19], or upregulation of the expression of LRP5/6 [20] or β-catenin [21] can also lead to the activation of β-catenin. CLA was shown to be a stimulator of Wnt3a and Wnt10b in the above studies; whether it also regulates GSK3β activity or targets the downstream of Wnt ligands is not clear. To elucidate this, dickkopf-1 (DKK1), an inhibitor of Wnt/β-catenin signaling transduction, was employed to block Wnt-initiated signaling transduction [5]. hBMSCs were induced for osteoblast differentiation with CLA (1 µM) alone, or in combination with DKK1 (0.5 µg/mL). It was found that addition of DKK1 successfully inhibited CLA-stimulated cellular ALP activity and extracellular calcium deposits (Figure 6). Consistent with this, CLA-promoted mRNA expression of *GSK3β*, *β-catenin*, *Runx2*, and *Osterix*, protein expression of phospho-GSK3β (Ser9), active β-catenin, Runx2, and OPN (Figure 7), and nuclear accumulation of β-catenin were also significantly inhibited by DKK1 (Figure 8). In this study, blocking the binding of Wnt ligands to their receptor successfully inhibited the CLA-promoted nuclear translocation of β-catenin and osteoblast differentiation, indicating that CLA did not target GSK3β, β-catenin, or other downstream members.

### 2.5. CLA-Promoted Osteoblast Differentiation Did Not Affect the OPG/RANKL Axis

The osteoprotegerin (OPG)/receptor activator of nuclear factor-κB ligand (RANKL) axis is crucial in the development of osteoporosis [22]. Whether CLA affects the OPG/RANKL axis or not was assessed by examining the mRNA levels of *OPG* and *RANKL*, the differentiating hBMSCs expressed, and the protein level of OPG and RANKL in the differentiating cells secreted into the media. Our results showed that treatment with CLA did not affect the mRNA expression of *OPG* and *RANKL* (Figure 9A,B) and the protein secretion of OPG and RANKL (Figure 9C,D), indicating an unaffected OPG/RANKL axis.

## 3. Materials and Methods

Melting point was measured on a Pyris Diamond Differential scanning calorimeter (Perkin Elmer, Waltham, UK). UV spectra were recorded on a TU-1901 UV spectrometer (Purkinje General, Bejing, China). IR spectra were collected on a Bruker Equinox 55 FTIR/FTNIR Spectrometer (Billerica, MA, USA). ESIMS data were obtained on an MDS Sciex API 2000 LC/MS/MS system (MDS Sciex, Ottawa, ON, Canada). High resolution (HR)-EIMS data were obtained on a MAT95XP mass spectrometer (Thermo Finnigan, Waltham, MA, USA). ^1^H NMR, ^13^C NMR, and heteronuclear multiple bond correlation (HMBC) spectra were recorded on a Bruker DRX-400 instrument using TMS as internal standard. Silica gel (200–300 meshes, Qingdao Marine Chemical Ltd., Qingdao, China) was used for column chromatography.

### 3.1. Reagents and Antibodies

The recombinant human DKK1 was purchased from PeproTech (Rocky Hill, NJ, USA). Primary antibodies for β-catenin, BMP-2, OPN, and collagen-1, horseradish peroxidase (HRP)-conjugated secondary antibodies (goat anti-rabbit or anti-mouse IgG H & L) and Alexa-488-conjugated secondary antibody were purchased from Abcam (Shanghai, China). Primary antibodies for GSK3β, phospho-GSK3β (Ser9), non-phospho(active)-β-catenin (Ser45), and Runx2 (O1L7F) were purchased from Cell Signaling Technology (Nanjing, China). DAPI was obtained from ZSGB-BIO (Beijing, China).

### 3.2. Isolation and Structure Identification of CLA

Air-dried leaves of *C. cajan* (5 kg, collected from Wenshan County, Yunnan Province, China) were extracted with 95% ethanol under reflux. The concentrated ethanol extract was scattered with 60 °C hot water (10 L), and placed overnight at room temperature (RT). After centrifugation, the precipitate was mixed with water (1 L) and sufficiently extracted with methylene chloride (CH_2_Cl_2_). The CH_2_Cl_2_-soluble extract (20 g) was subjected to silica gel columns and eluted with petrol ether (PE)–CH_2_Cl_2_ (50:50 to 10:90, *v*/*v*), and PE–acetone (90:10 to 80:20) to afford five fractions (A–E). Fraction A (5.1 g), an elution of PE–CH_2_Cl_2_ (50:50), was separated on a Silica gel column by eluting with PE–acetone (98:2) to provide **1** (300 mg, purity 99.0% by HPLC), which exhibited as colorless needles with melting point 154 °C. The structure of **1** was identified as cajanolactone A (CLA) (Figure 1A) [14]. Spectroscopic data of CLA: UV (MeOH) λ max (log ε) 205 (0.80), 265 (1.71), 319 (0.41), 351 (0.53) nm; IR (KBr) ν_max_ 3426, 2975, 1687, 1629, 1612, 1575, 1498, 1469, 1380 cm^−1^; ESIMS *m*/*z* 337.4 [M + H]^+^, 359.3 [M + Na]^+^, 673.6 [M + Na]^+^ and 695.5 [2M + Na]^+^; HR-EIMS *m*/*z* 336.1357 (calcd for C_21_H_20_O_4_, 336.1356). ^1^H NMR, ^13^C NMR, and HMBC spectral data are presented Table 1. For more details, please see Appendix A.

### 3.3. Cell Line and Cell Culture

hBMSCs were purchased from Cyagen Biosciences (Guangzhou, China) and grown in the growth medium (Cyagen Biosciences, Guangzhou, China) at 37 °C in a humidified atmosphere of 5% CO_2_ with a seeding density of 2 × 10^4^ cells per cm^2^. Cells were passaged or expanded by trypsinization when they grew to 80% confluence. The medium was replaced every 2 days during the incubation period. Cells from passages 4–6 were used in subsequent experiments.

### 3.4. Cell Proliferation Assay

hBMSCs were seeded into 96-well plates (5000 cells per well or 3000 cells per well in 100 μL growth medium), and allowed to adhere for 24 h, then cells were treated with different concentrations of CLA for 48 h (for the plate with 5000 cells per well) or 72 h (for the plate with 3000 cells per well) at the cell growth conditions. Cells without CLA treatment were set as control. Cell viabilities were measured using Cell Counting Kit 8 (Dojindo, Kumamoto, Japan) by following the manufacturer’s instruction. Effect of CLA on proliferation of hBMSCs was expressed as relative cell viability, by setting the viability of the control as 1.

### 3.5. Induction for Osteoblast Differentiation

hBMSCs were seeded (2 × 10^4^ cells per cm^2^) into 24-well plates in the growth medium and allowed to adhere for 24 h. After that, cells were induced for osteoblast differentiation in osteogenic differentiation medium (Cyagen Biosciences) for a period according to the needs of experiment. The media were replaced every 3 days. To investigate the effect of CLA on osteogenic differentiation, different concentrations of CLA (1, 2, 4 µM) were added to the differentiation medium from the start of osteogenic induction.

### 3.6. Detection of Osteoblast

Cellular ALP activity and extracellular calcium deposits are indicators of successful osteoblast differentiation. In this study, ALP activity and calcium deposits of differentiated osteoblasts were visualized by ALP activity staining kit and alizarin red S (ARS) staining kit (GenMed Scientifics Inc. USA, Shanghai, China) respectively, and taken pictures (100×) on a Leica DMI3000B inverted microscope (Leica Microsystems, Wetalar, Germany).

### 3.7. Quantitative Analysis of Cellular ALP Activity

Cells were lysed in radioimmunoprecipitation assay (RIPA) lysis buffer (Beyotime, Shanghai, China) and centrifuged at 12,000 rpm at 4 °C for 15 min. The supernatants were collected and ALP activities for different treatments were evaluated by employment of the ALP detection reagent kit (Beyotime). Experiments were performed in triplicates to reduce randomized error.

### 3.8. Quantitative Analysis of Extracellular Calcium Deposits

In a 24-well plate, extracellular calcium deposits of differentiated osteoblasts were stained with ARS as described above. Calcium-combined ARS was then dissolved with 1 mL 10% cetylpyridinum chloride (Cyagen Biosciences) for 15 min. Precisely 100 µL of the ARS-containing solution was transferred into a 96-well plate and the optical density was measured at 562 nm on a Synergy HT Multi-Mode Microplate Reader (BioTek Instruments Inc., Winooski, VT, USA). The calcium amount was directly proportional to the optical density of ARS.

### 3.9. Quantitative Analysis of Secreted OPG and RANKL

hBMSCs were induced for osteoblast differentiation as described above in Section 3.5. Media on the 1st day, 7th day, and 15th day of differentiation were collected. RANKL and OPG secreted into the media by differentiating cells were quantitatively analyzed using human OPG and RANKL ELISA kits (BOSTER, Wuhan, China).

### 3.10. RT-qPCR Analysis

hBMSCs were harvested at assigned time points post osteogenic induction. Total RNA was extracted using TRIzol Reagent (Life Technologies, Gaithersburg, MD, USA) according to manufacturer’s instruction. Reverse transcription was performed using the qPCR RT Kit (TOYOBO, Co. Ltd., Osaka, Japan) according to manufacturer’s instructions. mRNA levels of target genes were measured using quantitative PCR (qPCR) analysis with SYBR^®^ Green qPCR Mix (TOYOBO Co. Ltd., Osaka, Japan). The quantitative assay was performed in a 7500 Real-Time PCR system (Applied Biosystems; Thermo Fisher Scientific, Inc., Waltham, MA, USA) and the following PCR cycling protocol was used: 95 °C for 1 min, followed by 40 cycles of 95 °C for 15 s and 60 °C for 1 min. The relative expression of gene-specific products was analyzed using the 2^−△△CT^ method and normalized to the corresponding *GAPDH* values. The primer sequences used are shown in Table 2.

### 3.11. Western Blot Analysis

Cells with different treatments were lysed with lysis buffer containing 1 mM aprotinin, 1 mM pepstain, 1 mM NaF, 1 mM Na_4_P_2_O_7_, and 1 mM Na_3_VO_4_. Protein concentrations in the lysates were measured using the Enhanced BCA Protein Assay Kit (Beyotime, Shanghai, China). Twenty micrograms of the protein samples were subjected to SDS-PAGE on a 10% polyacrylamide gel and then transferred onto a polyvinylidene fluoride membrane (Millipore, Bedford, MA, USA). The membrane was incubated with corresponding primary antibody and HRP-labeled secondary antibody, successively. Subsequently, the proteins were visualized using an ECL kit (Millipore, Billerica, MA, USA). The immunoreaction signals were detected using the C-DiGit Blot Scanner system (LI-COR, Lincon, NE, USA). Protein quantification was performed using Image J v1.48 (National Institutes of Health, Bethesda, MD, USA).

### 3.12. Immunofluorescent Detection of β-Catenin

In 35 mm culture dishes with a round glass bottom (10 mm diameter), hBMSCs were induced for osteogenic differentiation for 48 h, in the absence or presence of 1 µM CLA alone, or 1 µM CLA in combination of 0.5 µg/mL DKK1. Cells were then fixed with 4% formaldehyde for 1 h at RT, permeabilized with 0.5% Triton X-100 in PBS for 30 min, and blocked with 5% bovine serum albumin for 30 min, successively. After that, cells were incubated with anti-β-catenin antibody (1:100) overnight at 4 °C, with goat anti-rabbit IgG conjugated to fluorescent Alexa-488 (1:100) for 2 h at RT, and then with DAPI to stain nuclei. Lastly, cells were mounted and observed under a confocal microscope (LSM 510 system, ZEISS International, Oberkochen, Germany).

### 3.13. Statistical Analysis

Data were expressed as mean ± SD of at least three repeated experiments. Differences between groups were analyzed using one-way analysis of variance, and *p* value < 0.05 was considered statistically different. Bar graphs were drawn using GraphPad Prism v6.0 software (GraphPad Software, Inc., San Diego, CA, USA).

## 4. Conclusions

Cajanolactone A, a stilbenoid discovered from *C. cajan*, promoted osteogenic differentiation of hBMSCs by stimulating Wnt3a and Wnt10, thus leading to activation of Wnt/LRP5/β-catenin signaling transduction. The present study suggests that CLA is a potential drug candidate for treatment of osteoporosis and fracture nonunion.

## Figures and Tables

**Figure 1 molecules-24-00271-f001:**
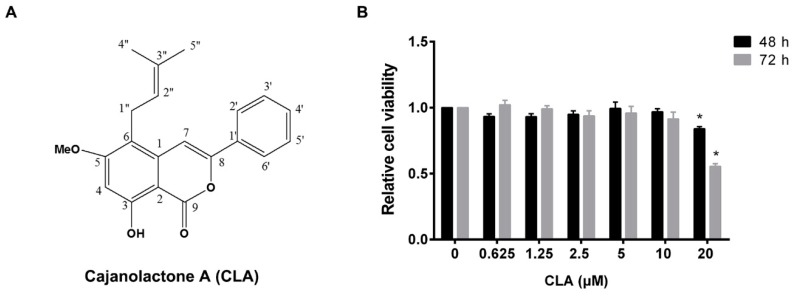
Effect of cajanolactone A (CLA) on proliferation of human bone marrow mesenchymal stem cells (hBMSCs). (**A**) Structure of CLA. (**B**) Relative viability of hBMSCs treated with various concentrations of CLA for 48 or 72 h. Data are expressed as mean ± SD (*n* = 3). Compared to control (0 μM), data were significantly different at * *p* < 0.05.

**Figure 2 molecules-24-00271-f002:**
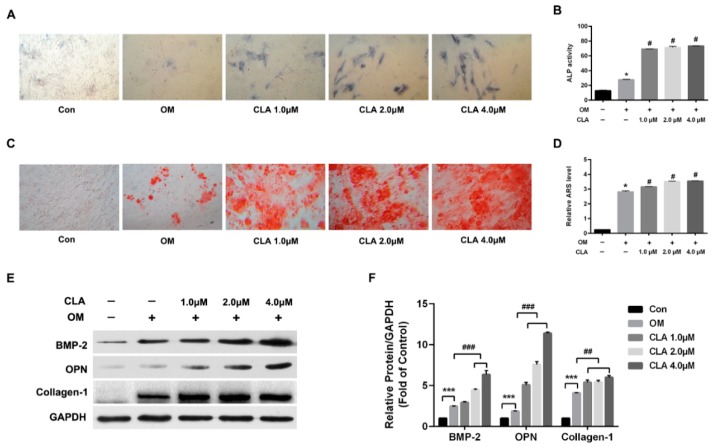
CLA promoted osteoblast differentiation in hBMSCs. hBMSCs (Con) were induced for osteoblast differentiation for 15 d, without (OM) or with different concentrations of CLA (CLA). (**A**) Cellular ALP activity (blue) detected using ALP activity staining kit (100×); (**B**) quantitative analysis of ALP activity; (**C**) extracellular calcium deposits (red) detected by alizarin red S (ARS) staining (100×); (**D**) quantitative analysis for calcium-combined ARS; (**E**) osteoblast-specific proteins detected by Western blotting; (**F**) protein expression levels normalized to GAPDH. Data are expressed as mean ± SD (*n* = 3). Compared to Con, data were significantly different at * *p* < 0.05 and *** *p* < 0.001; compared to OM, data were significantly different at ^#^
*p* < 0.05, ^##^
*p* < 0.01, ^###^
*p* < 0.001.

**Figure 3 molecules-24-00271-f003:**
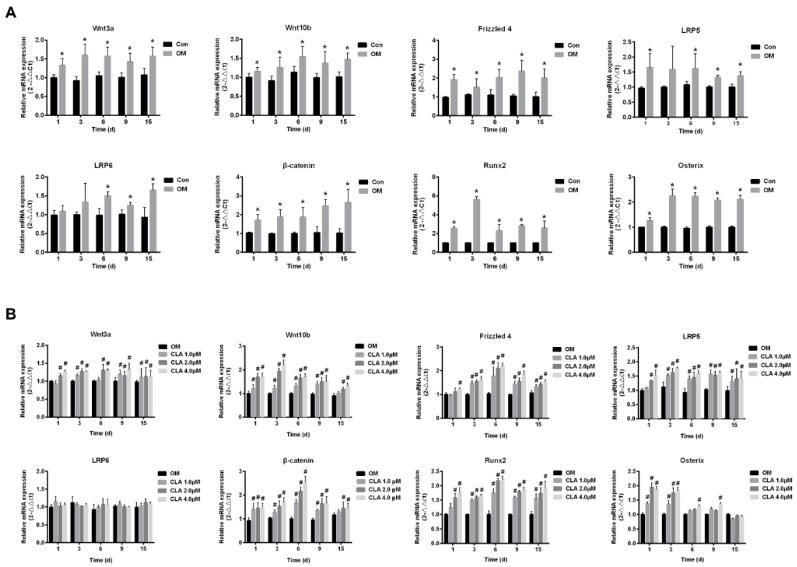
mRNA expression of Wnt/β-catenin signaling pathway-related genes in differentiating hBMSCs. hBMSCs were maintained in growth medium (Con), or induced for osteogenic differentiation for 1, 3, 6, 9, or 15 d respectively, in the absence (OM) or presence of CLA (CLA). mRNA levels of respective genes were measured by RT-qPCR and normalized to *GAPDH*. (**A**) Comparison of mRNA expression levels of respective genes in Con and OM. (**B**) Comparison of mRNA levels of respective genes in OM and CLA. Data were expressed as mean ± SD (*n* = 3). Compared to Con, data were significantly different at * *p* < 0.05; compared to OM, data were significantly different at ^#^
*p* < 0.05.

**Figure 4 molecules-24-00271-f004:**
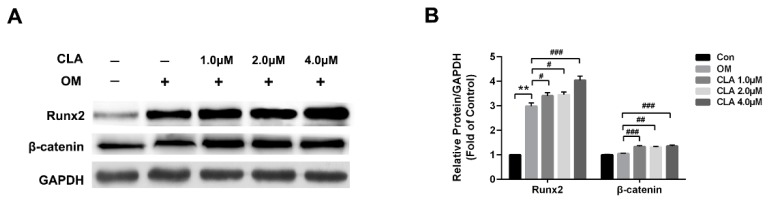
Expression of Runx2 and β-catenin in hBMSCs. Cells were maintained in growth medium, or induced for osteoblast differentiation in the presence or absence of different concentrations of CLA for 15 d. (**A**) Protein expression detected by Western blotting; (**B**) Quantitative analysis for protein expression. Compared to Con, data were significantly different at ** *p* < 0.01; compared to OM, data were significantly different at ^#^
*p* < 0.05, ^##^
*p* < 0.01, ^###^
*p* < 0.001.

**Figure 5 molecules-24-00271-f005:**
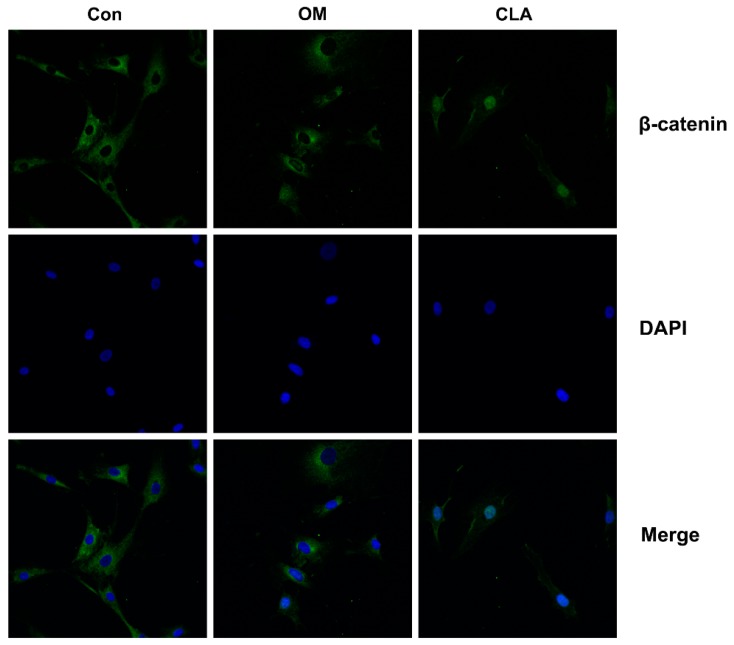
Confocal imaging for cellular β-catenin (100×). hBMSCs were maintained in growth medium (Con), or induced for osteoblast differentiation in the absence (OM) or presence of 1 µM CLA (CLA) for 48 h. Cellular β-catenin (green) was detected by immunofluorescence staining, with the nuclei stained with DAPI (blue). In Con, β-catenin expressed mainly in cytosol. In CLA, β-catenin expressed mainly in nucleus.

**Figure 6 molecules-24-00271-f006:**
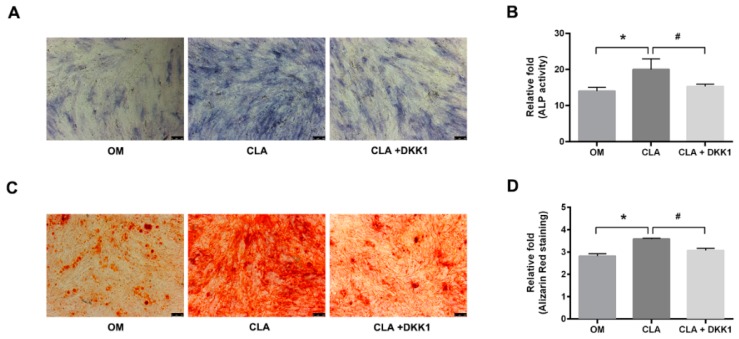
Effect of DKK1 on CLA-stimulated osteoblast differentiation. hBMSCs were induced for osteoblast differentiation for 15 d (OM), with 1 µM CLA alone (CLA), or 1 µM CLA in combination with 0.5 µg/mL DKK1 (CLA + DKK1). (**A**) Cellular ALP activity detected using ALP activity staining kit (100×); (**B**) cellular ALP activity quantified using the ALP detection reagent kit; (**C**) extracellular calcium deposits detected by ARS staining (100×); (**D**) quantitative analysis for calcium-combined ARS. Data are expressed as mean ± SD (*n* = 3). Compared to OM, data were significantly different at * *p* < 0.05; compared to CLA, data were significantly different at ^#^
*p* < 0.05.

**Figure 7 molecules-24-00271-f007:**
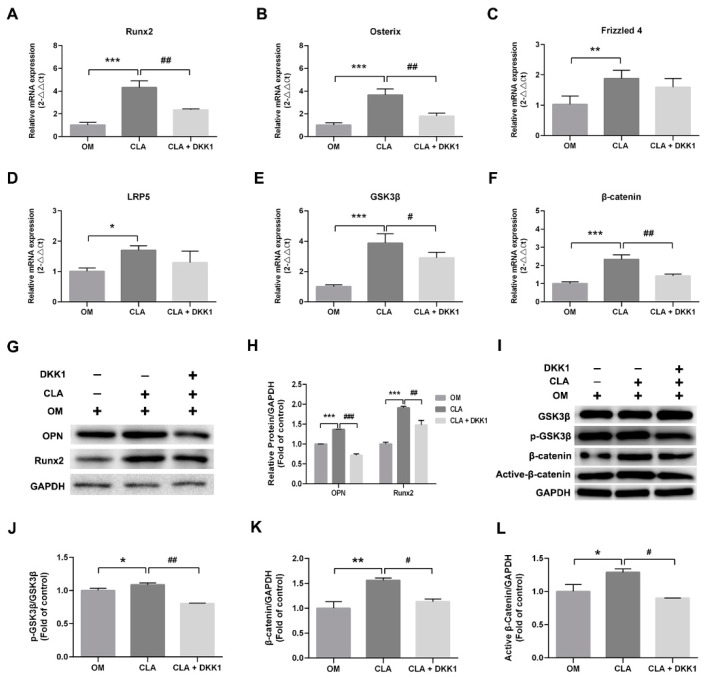
Effects of DKK1 on CLA-stimulated gene expression and protein expression in hBMSCs. Cells were induced for osteoblast differentiation for 7 d (OM), with 1 µM CLA alone (CLA), or with 1 µM CLA and 0.5 µg/mL DKK1 (CLA + DKK1). (**A**–**F**) RT-qPCR analysis for mRNA expression of *Runx2* (**A**), *Osterix* (**B**), *Frizzled 4* (**C**), *LRP5* (**D**), *GSK3β* (**E**), and *β-catenin* (**F**). (**G**) Western blot analysis for protein expression of Runx2 and OPN, and (**H**) protein expression of Runx2 and OPN normalized to GAPDH; (**I**) Western blot analysis for protein expression of GSK3β, p-GSK3β (Ser9), β-catenin and active-β-catenin; (**J**–**L**) quantified protein expression normalized to GAPDH. Data are expressed as mean ± SD (*n* = 3). Compared to OM, data were significantly different at * *p* < 0.05, ** *p* < 0.01, *** *p* < 0.001; compared to CLA, data were significantly different at ^#^
*p* < 0.05, ^##^
*p* < 0.01, ^###^
*p* < 0.001.

**Figure 8 molecules-24-00271-f008:**
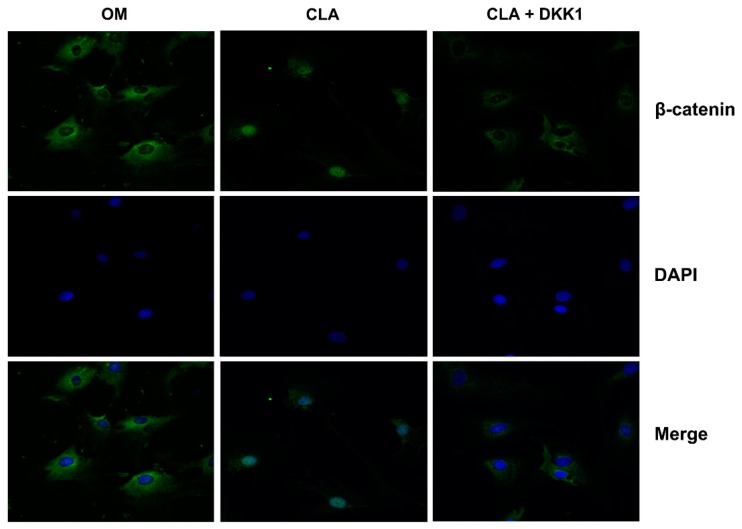
DKK1 inhibited CLA-stimulated nuclear translocation of β-catenin (100×). hBMSCs (Con) were induced for osteogenic differentiation for 48 h (OM), with 1 µM CLA alone (CLA), or 1 µM CLA and 0.5 µg/mL DKK1 (CLA + DKK1). Expression of β-catenin (green) was detected by immunofluorescence staining, with nuclei stained using DAPI (blue).

**Figure 9 molecules-24-00271-f009:**
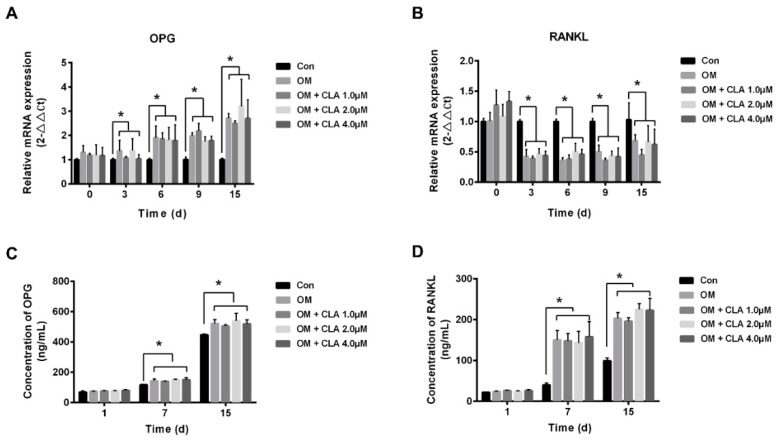
Effect of CLA on OPG/RANKL axis. hBMSCs were maintained in growth medium (Con), or induced for osteogenic differentiation for 1, 3, 6, 9, or 15 d respectively, in the absence (OM) or presence of CLA (CLA). mRNA levels of *OPG* (**A**) and *RANKL* (**B**) were measured by RT-qPCR and normalized to *GAPDH*; protein levels of OPG (**C**) and RANKL (**D**) in culture media were measured using ELISA kits. Data were expressed as mean ± SD (*n* = 3). Compared to Con, data were significantly different at * *p* < 0.05.

**Table 1 molecules-24-00271-t001:** ^1^H NMR, ^13^C NMR, and HMBC data of CLA measured in CDCl_3_ (δ ppm).

Position	δ_H_ ^1^ (J in Hz)	δ_C_ ^2^, Type	HMBC Correlations ^3^ (H to C)
1		136.0, C	1″
2		99.6, C	4, 7, OH
3		162.4, C	4
4	6.52 s	98.2, CH	
5		164.3, C	4, 1″, OMe
6		116.0, C	4, 7, 1″
7	7.04 s	100.0, CH	
8		152.0, C	7, 2′, 6′
9		166.3, C	
2′, 6′	7.82 d (8.0)	125.2, CH	2′, 3′, 4′, 5′, 6′
3′, 5′	7.44 m	128.8, CH	4′
4′	7.44 m	129.9, CH	2′, 3′, 5′, 6′
1″	3.45 br d (4.0)	23.6, CH2	2″
2″	5.05 br s	122.6, CH	4″, 5″
3″		131.9, C	1″, 4″, 5″
4″	1.68 s	25.7, CH3	2″, 5″
5″	1.84 s	17.9, CH3	2″, 4″
5-OMe	3.89 s	55.9, CH3	
3-OH	11.28 s		

Note: ^1^ Measured at 400 MHz. ^2^ Measured at 100 MHz. ^3^ HMBC correlations: H to C. Assignments were supported with HMBC spectra.

**Table 2 molecules-24-00271-t002:** Primer sequences for RT-qPCR.

Genes	Primer Sequence (5′–3′)	
	Forward	Reverse
*Wnt3a*	TGGTGTCTCGGGAGTTCGC	CCGTGGCACTTGCACTTGA
*Wnt10b*	CTCCTGTTCCTGGCGTTGTG	GCAACTTCAGGCCCAGAATC
*LRP5*	GCAGGTCTTCATGTCACTCAGCAG	TCAAAGCTGTGAATGTGGCCAAGG
*LRP6*	GCAGCCTGTGGGACTTACTGTGTT	TGAGCACAAGGGTGCTGTCTGTAT
*GSK3β*	GGCAGCATGAAAGTTAGCAGA	GGCGACCAGTTCTCCTGAATC
*Runx2*	GTTCAACGATCTGAGATTTG	GGGGTCTGTAATCTGACTCT
*Osterix*	ACCCAACATCAAGCAGAACCAC	TCTTCAACTTCCCAGGCTCACT
*β-catenin*	AGCTTCCAGACACGCTATCAT	CGGTACAACGAGCTGTTTCTAC
*Frizzled 4*	AAGGATGGGACAAAGACAGACAA	ACATGCCTGAAGTGATGCCCA
*RANKL*	GGAAAGAAAGTGGGAGCAGA	ATTAGGCCCTTCAAGGTGTC
*OPG*	TCTGGTTCCCATAAAGTGAGTC	CGAAAGCAAATGTTGGCATA
*GAPDH*	TCAACGACCACTTTGTCAAGC	TGTGGGCCATGAGGTCCACCA

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
