# Peer review of "Cajanolactone A from Cajanus cajan Promoted Osteoblast Differentiation in Human Bone Marrow Mesenchymal Stem Cells via Stimulating Wnt/LRP5/β-Catenin Signaling"

_molecules, 2019, doi:10.3390/molecules24020271_

Reviewer 1 Report

The paper presented by Liu and colleagues "Cajanolactone A from Cajanus cajan promoted osteoblast differentiation in human bone marrow mesenchymal stem cells via stimulating Wnt/LRP5/b-catenin signaling" describes the effect of a natural stilbenoid cajanolactone A. CJA, on osteogenic differentiation of bone marrow mesenchymal cell. Data showed CJA activates wnt/beta catenin canonical pathway and its action is mitigated by the soluble antagonist of wnt, dkk1. The paper presents some interesting data, but some questions arise to better understand the manuscript.

1) in Fig 1 structure of CLA is presented. Have the authors some information regarding the Structure-Activity-Relationship?

2) in Fig 2 cell viability is reported, noting a reduction at 20microM both at 48 and 72h. Since the assay measures the enzymatic activity of cells, the reported reduction should be linked to a steady-state due to the triggering of a differentiation process and the arrest of the cell cycle rather than a toxic effect of CLA. Authors should provide further analysis such cell cycle or apoptotic marker, or change accordingly the sentence.

3) In fig 2E analysis of osteogenic markers are reported. Have the authors looked at RANKL/OPG axis, that is crucial for osteoporosis onset?

4) The authors investigated the canonical WNT/beta catenin pathway. Have they any results for non.canonical (i.e. calcium oscillation) and planar cell polarization (JNK/Rho/Ras)pathways?

5) Regarding the canonical WNT pathway, dkk1 is used to antagonize the effect of CLA, concluding that CLA exerts effects upstream in the pathway. However is not clear how CLA acts on the receptors. Authors should investigate whether CLA direct effect or synergic to WNT3 on Frizzled 4 activation. Silencing by SiRNA of shRNA of endogenous wnt3 and treatment with CLA assessing to activity of dishevelled activation could help to assess this issue. 

6) Moreover, a dose-dependent treatment with dkk1 should be performed to reveal if the inhibition is a competitive or non-competitive one.

7) CLA has similarity to resveratrol, a stilbene with anti-cox2 activity. Cox2 is known to regulate wnt signaling. Can the authors exclude that the effects of CLA are not dependent on COX2 regulation rather than a direct effect on wnt pathway?.

Minor concerns:

8) FIG 7I: p-GSK3b is very weird, please provide a better exposition.

9) Many typo and spelling errors are present. Please check carefully

Author Response

Response to reviewer

1)       in Fig 1 structure of CLA is presented. Have the authors some information regarding the Structure-Activity-Relationship?

Response: A number of stilbenoids had been reported to promote osteogenesis and inhibit adipogenesis. Osteogenic promoting activity of CLA should also be attributed to its stilbene structure. However, structure-activity relationship of CLA is unknown, and it would be our future research interest.

2)       in Fig 2 cell viability is reported, noting a reduction at 20microM both at 48 and 72h. Since the assay measures the enzymatic activity of cells, the reported reduction should be linked to a steady-state due to the triggering of a differentiation process and the arrest of the cell cycle rather than a toxic effect of CLA. Authors should provide further analysis such cell cycle or apoptotic marker, or change accordingly the sentence.

Response: Thank reviewer’s advice. In revised manuscript, “no-effect concentration” was used to replace “non-toxic concentration” as advised.

3)       In fig 2E analysis of osteogenic markers are reported. Have the authors looked at RANKL/OPG axis, that is crucial for osteoporosis onset?

Response: Yes, we had checked the mRNA expression and protein secretion of OPG and RANKL indeed. CLA exhibited no effect on OPG and RANKL. In revised manuscript, this part of work was added (subtitle 2.5. and 3.9.).

4)       The authors investigated the canonical WNT/beta catenin pathway. Have they any results for non.canonical (i.e. calcium oscillation) and planar cell polarization (JNK/Rho/Ras) pathways?

Response: No, we have not.

5)       Regarding the canonical WNT pathway, dkk1 is used to antagonize the effect of CLA, concluding that CLA exerts effects upstream in the pathway. However is not clear how CLA acts on the receptors. Authors should investigate whether CLA direct effect or synergic to WNT3 on Frizzled 4 activation. Silencing by SiRNA of shRNA of endogenous wnt3 and treatment with CLA assessing to activity of dishevelled activation could help to assess this issue. 

Response: Thank reviewer’s advice. We had tried to elucidate if CLA also regulates LRP5 or Frizzled 4, by employment of Wif1 which is a Wnt inhibitor that directly binds to Wnts. Unfortunately the experiment failed: the first test suffered a power cut, and the second test a lab contamination, with the Wif1 and hBMSCs run out. We understand that further investigation is necessary for exploration of CLA – Wnt receptor relationship, but our funding is not sufficient to support the experiment the reviewer advised. In order to avoid ambiguity, the title “2.4. CLA did not target the downstream of Wnt ligands” was revised as “2.4. CLA did not target the downstream of canonical Wnt pathway” in the newly submitted manuscript.

6)       Moreover, a dose-dependent treatment with dkk1 should be performed to reveal if the inhibition is a competitive or non-competitive one.

Response: Thank reviewer’s advice. As mentioned above, our funding is not sufficient to support more experiments.

7)       CLA has similarity to resveratrol, a stilbene with anti-cox2 activity. Cox2 is known to regulate wnt signaling. Can the authors exclude that the effects of CLA are not dependent on COX2 regulation rather than a direct effect on wnt pathway?

Response: In our study, Cox2 or the crosstalk between Cox2 and Wnt/β-catenin signaling pathway had not been investigated. From our results we cannot exclude that the effects of CLA are not dependent on COX2 regulation.

Minor concerns:

8) FIG 7I: p-GSK3b is very weird, please provide a better exposition.

Response: Done as advised.  

9) Many typo and spelling errors are present. Please check carefully

Response: Done as advised.

Reviewer 2 Report

Manuscript Number: molecules-418925

The manuscript entitles « Cajanolactone A from Cajanus cajan promoted osteoblast differentiation in human bone marrow mesenchymal stem cells via stimulating Wnt/LRP5/b-catenin signaling » by Sham Liu et al., presents original data on the pro-differentiation effect of Cajanolactone A on human bone marrow mesenchymal stem cells.

The data reports in the manuscript are very interesting however comments need to be addressed to definitively persuade the reviewer.

Comments:

The CLA effects on proliferation and differentiation were analyzed at different times!

How can authors exclude an impact of CLA on proliferation after 15 days of treatment?

Some blots in Figure7I are saturated! Have to be changed!

English has to be improved in some parts and orthographic errors are present for instance line-18 “Mechanistic” and line-56 “mesenchymal”.

References 1 and 2 are not in the text!

The reviewer would appreciate the comments to be considered for a revised version of this manuscript.

Author Response

Response to Reviewer

Comments:

The CLA effects on proliferation and differentiation were analyzed at different times!

Response: Yes. The purposes of proliferation assay were to (1) investigate if CLA stimulates the proliferation of hBMSCs and (2) determine the working concentrations of CLA in followed differentiation experiments. Our results showed that, CLA did not stimulate the proliferation of hBMSCs. The maximum no-effect concentration on cell proliferation was 10 µM. Based on this, maximum working concentration of CLA in followed differentiation experiments were set as 4µM(2.5 times smaller than 10µM), in order to avoid to affect cell viability.

How can authors exclude an impact of CLA on proliferation after 15 days of treatment?

 Response: It is a very good question. Based on the result from a 72 h assay, we cannot conclude that CLA (4 µM) would not affect the proliferation of hBMSCs in a 15 d assay. Fortunately in the differentiation experiment, CLA (4, 2, 1 µM) dose-dependently enhanced osteoblast differentiation in hBMSCs, we can deduced that CLA at its working concentrations did not affect the viability of differentiating hBMSCs.

Some blots in Figure7I are saturated! Have to be changed!

Response: We are sorry. The old one had been replaced with a new one with shorter exposure time. 

English has to be improved in some parts and orthographic errors are present for instance line-18 “Mechanistic” and line-56 “mesenchymal”.

 Response: Thank reviewer. English has been improved and the orthographic errors have been corrected as advised.

References 1 and 2 are not in the text!

 Response: Forgive us for our negligenceErrors have been corrected in the new submission.

Round  2

Reviewer 1 Report

The Authors deal with reviewer's comments and provide many required experiments during the revision. The final version is adequate for publication.